# Women's experiences and outcomes of abortion care in sub-Saharan countries: A mixed methods systematic review protocol

Negash Wakgari [1,2]*, Gizachew A. Tessema[3,4], Stuart J. Watson[1,5,6], Delayehu Bekele[7], Zoe Bradfield[1,5]

1 Curtin School of Nursing, Curtin University, Perth, Western Australia, Australia, 2 Department of Midwifery, College of Medicine and Health Sciences, Ambo University, Ambo, Ethiopia, 3 Curtin School of Population Health, Curtin University, Perth, Western Australia, Australia, 4 enAble Institute, Curtin University, Perth, Western Australia, Australia, 5 Women and Newborn Health Service, North Metropolitan Health, Perth, Western Australia, Australia, 6 Psychology, Murdoch University, Perth, Western Australia, Australia, 7 Department of Obstetrics and Gynaecology, Saint Paul's Hospital Millennium Medical College, Addis Ababa, Ethiopia

* negashwakgari@yahoo.com

**Data Availability Statement:** No datasets were generated or analysed during the current study. All

## Abstract

### Introduction

Abortion care experiences encompass various aspects, including women's decision-making capability, physical and emotional experiences, service provision, and post-abortion experiences. The lack of woman-centred and respectful abortion services, influenced by stigma and restrictive abortion laws in certain contexts, poses a public health concern. These challenges may lead to variations in women's experiences and care outcomes, potentially resulting in adverse physical, psychological, and emotional outcomes for individuals seeking abortions. Therefore, this systematic review aims to synthesise the available evidence on women's abortion care experiences and outcomes in sub–Saharan Africa published from 2010 onwards.

### Methods

Eight databases including Medline, Embase, Scopus, CINAHL, Cochrane Library, PsychInfo, Web of Science, and Global Health will be searched using subject headings and specific keywords related to women's abortion care experiences, abortion care outcomes, and its measurement. Predetermined criteria will be used to select studies that meet the review's inclusion criteria. These include all original studies published in English languages that focussed on induced abortion care and assessed women's abortion care experiences and outcomes. After screening for title and abstract and full text, included studies will undergo data extraction, where information relevant to the methodological quality of each study will be collected. This review will integrate qualitative and quantitative data through a narrative synthesis approach.

relevant data from this study will be made available upon study completion.

**Funding:** The author(s) received no specific funding for this work.

**Competing interests:** The authors have declared that no competing interests exist.

**Abbreviations: JBI**, Joanna Briggs Institute; **PCC**, Population, Concept, and Context; **RISMA-P**, Preferred Reporting Items for Systematic Reviews and Meta-Analysis Protocols; **SSA**, Sub-Saharan Africa.

## Discussion

By synthesising abortion care experiences and outcomes across studies and analysing the commonalities and differences of the multifactorial challenges women face in health facilities, this study will improve the understanding of abortion care experiences and outcomes and inform evidence-based recommendations and future research directions. In addition, this systematic review will also discover and locate an existing measurement tool for abortion care experiences and outcomes for women while receiving the services in the facility.

## Introduction

Abortion remains a public health concern affecting women's reproductive health rights [1, 2], attributing to substantial morbidity and mortality in resource-limited countries. Sub-Saharan Africa (SSA) shares high burdens of abortion-related morbidity and mortality [3–5], contributing 70% of global maternal deaths in 2020 [6]. From 2010–2014, 77% of abortions were estimated to be unsafe in SSA, while the proportions were estimated to be 45% worldwide [3]. The disproportionate burden is associated with the lack of service availability and accessibility, restricted abortion policy, poor infrastructure, lack of skilled providers, abortion-related stigma and discrimination in the region [3, 7–11], exacerbating abortion care experiences and outcomes.

Women's experiences of abortion care may be varied and include experiences related to women's decision-making, the access and provision of abortion service-related facilities, and post-abortion [11–13]. The lack of woman-centred and respectful abortion services remains a public health concern, partly due to its associated stigma and restrictive abortion laws in some contexts [1, 14]. This may attribute to variations in women's experiences and care outcomes, including poor health outcomes for individuals seeking abortions as they may face barriers to accessing and receiving timely and appropriate medical care [15–17].

Experiences of abortion care can vary widely based on factors such as the legal context, availability of services, societal norms, and individual circumstances. Some evidence shows these experiences may include long waiting times to receive services, perceived discrimination and stigma, the ability to receive woman-centred services, client-provider interaction including communication, counselling, supportive care, informed decision-making, privacy, and confidentiality [18–22]. Moreover, abortion care outcomes may include potential moral injuries as women living in restrictive abortion settings may be required or encouraged to change their report of pregnancy circumstances to fit narrow criteria for abortion eligibility. Further outcomes include concerns for future pregnancy, anxiety, sadness, perceived guilt, and shame, and losing faith, and perceived relief and happiness [23–25].

The Maputo protocol, which emphasised equality and women's reproductive health rights in Africa, was adopted in 2003 and effective in 2005, and a total of 43 African nations signed the protocol in 2007 [26]. Maputo protocol article 14 described women's sexual and reproductive health rights, authorising women to access abortion services in cases of sexual violence, rape, incest, and where the continued pregnancy endangers the life of the mother or the foetus. This period was the time when some sub-Saharan countries started reforming their abortion policies, developing abortion policies implementation guidelines and expanding access to safe abortion services [27–31]. Our systematic review will be focused on studies published from 2010 onwards, seeing the recent developments in abortion care practices and legal changes that influence abortion care experiences and outcomes in SSA.

Research on abortion care experiences and outcomes is especially important in SSA, where access to safe abortion care is often limited and women may face various barriers to obtaining the care, they need [7, 15, 29, 30, 32–36]. There has been limited work to synthesise providers' [37, 38] and women's experiences of abortion care worldwide [39, 40]. Although understanding providers' experiences is compelling, a comprehensive understanding of abortion care requires considering the experiences, perspectives, and needs of the women themselves. However, there has not been a systematic assessment of the existing literature on abortion care experiences and outcomes, specifically from women's perspectives in SSA. Hence, this review will synthesise the available evidence base to collate the abortion care experiences and outcomes, which will help inform programs and healthcare provisions in abortion care services in SSA. This review also discovers the existing instruments for abortion care experiences and outcomes from women's perspectives.

## The review questions

1. What are women's experiences of abortion care and outcomes in SSA countries?

2. What measurement tool exists to measure women's abortion care experiences and outcomes?

## Methods

### Study design

This systematic review protocol was registered in PROSPERO (ID: CRD42023461963). We will use the Joanna Briggs Institute (JBI) approach to Mixed Methods Systematic Reviews [41]. The approach to integrating both qualitative and quantitative data will be adopted to ensure a comprehensive understanding of the multifactorial challenges women face while receiving abortion care. The Preferred Reporting Items for Systematic Reviews and Meta-Analysis Protocols (PRISMA-P) were used to define and develop the key content of a review protocol [42]. We will use the PRISMA 2020 guideline for reporting the results of systematic reviews [43].

### Inclusion and exclusion criteria

Studies including quantitative such as cross-sectional, case-control and cohort studies, qualitative, and mixed research that assessed women's experiences in abortion care and abortion care outcomes in SSA from 1 January 2010 onwards will be included. Studies focusing on induced abortion care services (pre-abortion care, receipt of abortion care, and post-abortion care) will be included. The Population, Concept, and Context (PCC) framework was adopted to structure review questions and guide the search for evidence (Table 1). Studies published in non-English languages, articles focusing on technical aspects of providing abortion care, such as studies focusing on the effectiveness of surgical or medication abortions, scoping reviews, systematic reviews, case reports, case series will be excluded. Similarly, studies focusing on the quality of abortion care, miscarriage, or spontaneous abortion care will also be excluded.

### Search strategy

First, a preliminary search in specific databases (Medline and CINAHL) was performed to help identify relevant articles and provide a foundation for developing a more comprehensive search strategy. During the limited exploration, free text terms and keywords found in the titles and abstracts of the relevant articles to the research topic were used to develop a full search

**Table 1. The PCC framework for identifying the eligibility of the studies for the research question, 2024.**

| Criteria | Element(s) | Descriptions |
|---|---|---|
| Population | Women | All women seeking induced abortion care services |
| Concept | Abortion care experiences | Any experiences, favourable or unfavourable, women face in health facilities while receiving induced abortion services. These may include long waiting times to receive services, perceived discrimination and stigma, the ability to receive woman-centred services, client-provider interaction, including communication, counselling, supportive care, informed decision-making, privacy, and confidentiality. |
| | Abortion care outcomes | Abortion care outcomes may include healthcare intervention outcomes, and any possible moral injuries reported from the studies, including concerns for future pregnancy, anxiety, sadness, perceived guilt and shame, loss of faith, perceived relief and happiness, care satisfaction, perceived pain more than expected, failed medication abortion, heavy bleeding, and infection. |
| | Measurement tool | Any tool exists to measure women's abortion care experiences and outcomes |
| Context | Sub-Saharan Africa | Any country in sub-Saharan Africa that reported on abortion care experiences and outcomes and its measurement tool |

strategy for Medline, Scopus, Embase, Web of Science, CINAHL, Cochrane Library, Psych-Info, and Global Health. We used Medical Subject Headings (MeSH), keywords, and free text search terms. Within the search terms, the full search strategy will include alternative terms for abortion care experiences and outcomes and combine them using different Boolean operators. In addition, snowballing will be applied to screen the references of identified articles for potentially relevant studies. An example of the entire search strategy to be undertaken is comprised in Table 2. The search scheme in table two has been prepared for use in the Medline database and will be fitted to relevant databases.

## Data screening

Once all relevant databases have been searched, citations will be exported to Covidence to streamline the screening process. Two reviewers will independently assess the eligibility of studies based on predetermined inclusion and exclusion criteria. Initial screening (round 1) will be done based on the title and abstract of each study. The remaining studies will be subjected to a full-text review (round 2) to determine their eligibility for inclusion. In cases of disagreement between the two reviewers during the screening process, a third reviewer will be involved to reach a consensus. The reasons for excluding studies during the full-text review (round 2) will be documented [44]. All studies reached a consensus by two reviewers, passed the full-text reviews, and will be subjected to data extraction.

## Quality assessment

The JBI critical appraisal checklists will be used to assess and evaluate the quality and rigor of the included studies. For quantitative studies (analytical cross-sectional, cohort, and case-control) and quantitative components of mixed methods studies, the JBI critical appraisal instrument will be used to evaluate different aspects of a study, such as a study design, sample size, data collection methods, and statistical analysis [45]. Similarly, the qualitative studies will be assessed by JBI checklists prepared for qualitative data [46]. Regardless of the quality assessment results, all studies that fulfil predetermined inclusion criteria will be included in the review and undergo synthesis. However, the critical appraisal results will be presented in tabular form with systematic review findings.

## Data extraction

A data extraction form will be adapted from the JBI manual for evidence synthesis. The JBI quantitative data extraction tool [47] and the JBI qualitative data extraction tool [48] will be

**Table 2. Search strategy for Medline databases, 2024.**

| Search # | Search string |
|---|---|
| #1 | exp "Abortion, Induced"/ OR exp "Abortion Applicants"/ |
| #2 | (Abort* OR postabortion OR "post-abort*").ti,ab. |
| #3 | (Pregnanc* ADJ3 (terminat* or end*)).ti,ab. |
| #4 | #1 OR #2 OR #3 |
| #5 | exp "Patient Satisfaction"/ OR exp "Social Stigma"/ OR exp "Social Discrimination"/ OR exp "Social Norms"/ OR exp "Social Perception"/ OR exp "Decision Making"/ OR exp "Emotions"/ OR exp "Depression"/ OR exp "Stress, Psychological"/ OR exp "Stress Disorders, Post-Traumatic"/ |
| #6 | (Satisf* OR care OR access* OR service* OR patient-centred care OR person-centred care OR communicat* OR delay OR wait* OR experienc* OR stigma* OR discriminat* OR norms OR perception* OR decision* normal* OR justice OR law OR restrict*).ti,ab. |
| #7 | (Emotion* OR fear OR worry OR pain OR sad* OR guilt* OR shame OR embarrass* OR faith OR relief OR happy* OR anxiety OR anxious* OR depress* OR stress* OR outcome* OR consequence* OR pain OR psychosocial OR "psycho-social" OR "moral injur*").ti,ab. |
| #8 | (Measure* OR tool OR instrument* OR "scale").ti,ab. |
| #9 | #5 OR #6 OR #7 OR #8 |
| #10 | exp "Africa South of the Sahara"/ |
| #11 | ("sub-Saharan Africa*" OR Angola OR Benin OR Botswana OR "Burkina Faso" OR Burundi OR "Cabo Verde" OR Cameroon OR "Central African Republic" OR Chad OR Comoros OR Congo OR "Cote d'Ivoire" OR Djibouti OR "Equatorial Guinea" OR Eritrea OR Eswatini OR Ethiopia OR Gabon OR Gambia OR Ghana OR Guinea OR "Guinea-Bissau" OR Kenya OR Lesotho OR Liberia OR Madagascar OR Malawi OR Mali OR Mauritania OR Mauritius OR Mozambique OR Namibia OR Niger OR Nigeria OR Rwanda OR "Sao Tome and Principe" OR Senegal OR Seychelles OR "Sierra Leone" OR Somalia OR "South Africa" OR "South Sudan" OR Sudan OR Tanzania OR Togo OR Uganda OR Zambia OR Zimbabwe).ti,ab. |
| #12 | #10 OR #11 |
| #13 | #4 AND #9 AND #12 |

used separately to collect relevant information from each reviewed study. Two reviewers will independently extract data from included studies. Relevant data, such as information about the first author's/s' name/s, publication year, aim of the study, study design, sample size, methodology, and key findings, will be extracted from the selected studies. The extracted data will be compared to ensure consistency in how the data is being interpreted and collected. Discrepancies among reviewers will be addressed through discussion or with a third reviewer. When reviewers disagree on certain aspects of a paper, the first step is typically to engage in discussion to address the disagreements. If disagreements persist after the initial discussion, a third reviewer can provide an impartial perspective and help mediate the discussion between the initial reviewers. Further, if reviewers identify gaps in the data presented in a paper, they may contact the authors to request additional information or clarification for the missing data [49].

## Data synthesis and integration

The extracted data will be checked and summarised to address the review question. The included studies' characteristics and data related to the review questions will be organised using tables and narrative summaries under abortion care experiences, abortion care outcomes, and measurement tools for abortion care experiences and outcomes. A narrative synthesis to organise and analyse the data will be employed. Qualitative and quantitative data will be analysed separately and presented in a narrative format. The quantitative and qualitative data included in this review will then be organised according to the JBI methodology for mixed methods systematic review [49]. Then, a convergent, segregated approach will be used

to synthesise and integrate the data This integration can occur during the interpretation of results or when drawing conclusions.

## Ethics approval

Ethical approval is not required since this review will employ published articles. Given that included studies will be presenting anonymised data, we do not anticipate any issue pertaining to ethical implications.

## Discussion

The main aim of this review is to synthesise evidence on women's abortion care experiences and outcomes in SSA. Abortion care experiences and outcomes in SSA can vary widely due to the diverse social, cultural, legal, and healthcare contexts within the region. Access to safe abortion services is limited in many sub-Saharan African countries, contributing to unsafe abortion practices and associated health risks [7, 50, 51]. By synthesising abortion care experiences and outcomes across studies and analysing the commonalities and differences of the multifactorial challenges women face in health facilities, from admission to discharge, this study will improve the understanding of abortion care experiences and outcomes in the region. In addition, this systematic review will also discover and locate an existing measurement tool for abortion care experiences and outcomes for women while receiving the services in the facility. These will help researchers to develop standardised measurement tools for abortion care experiences and outcomes that will help policymakers, program managers, and abortion providers to monitor women's abortion care experiences and outcomes and subsequently improve the quality of abortion care in SSA. Overall, the findings of this review will identify aspects of abortion care experiences that women may encounter in facilities and care outcomes that could directly influence abortion care policies and practices in SSA.

## Supporting information

**S1 Checklist. PRISMA-P (Preferred Reporting Items for Systematic review and Meta-Analysis Protocols) 2015 checklist: Recommended items to address in a systematic review protocol\*.**
(DOCX)

## Acknowledgments

We appreciate the contribution of Vanessa Varis, a faculty librarian at the Faculty of Health Sciences at Curtin University, who commented on the drafts of the search strategy.

## Author Contributions

**Conceptualization:** Negash Wakgari, Gizachew A. Tessema, Stuart J. Watson, Zoe Bradfield.

**Investigation:** Negash Wakgari, Gizachew A. Tessema, Stuart J. Watson, Zoe Bradfield.

**Methodology:** Negash Wakgari, Gizachew A. Tessema, Stuart J. Watson, Zoe Bradfield.

**Project administration:** Gizachew A. Tessema, Stuart J. Watson, Delayehu Bekele, Zoe Bradfield.

**Supervision:** Gizachew A. Tessema, Stuart J. Watson, Delayehu Bekele, Zoe Bradfield.

**Validation:** Gizachew A. Tessema, Stuart J. Watson, Delayehu Bekele, Zoe Bradfield.

**Writing – original draft:** Negash Wakgari.

**Writing – review & editing:** Negash Wakgari, Gizachew A. Tessema, Stuart J. Watson, Delayehu Bekele, Zoe Bradfield.

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
