## [Decision Letter · Decision Letter 0]

2 Sep 2024

PONE-D-24-05732Women's Experiences and Outcomes of Abortion Care in Sub-Saharan Countries: A Mixed Methods Systematic Review ProtocolPLOS ONE

Dear Dr. Wakgari,

Thank you for submitting your manuscript to PLOS ONE. After careful consideration, we feel that it has merit but does not fully meet PLOS ONE’s publication criteria as it currently stands. Therefore, we invite you to submit a revised version of the manuscript that addresses the points raised during the review process.

We look forward to receiving your revised manuscript.

Kind regards,

Sisay Abebe Debela

Academic Editor

PLOS ONE

Journal Requirements:

Reviewers' comments:

Reviewer's Responses to Questions

**Comments to the Author**

1. Does the manuscript provide a valid rationale for the proposed study, with clearly identified and justified research questions?

Reviewer #1: Yes

Reviewer #2: Yes

Reviewer #3: Yes

2. Is the protocol technically sound and planned in a manner that will lead to a meaningful outcome and allow testing the stated hypotheses?

Reviewer #1: Yes

Reviewer #2: Yes

Reviewer #3: Yes

3. Is the methodology feasible and described in sufficient detail to allow the work to be replicable?

Reviewer #1: Yes

Reviewer #2: Yes

Reviewer #3: Yes

4. Have the authors described where all data underlying the findings will be made available when the study is complete?

Reviewer #1: No

Reviewer #2: Yes

Reviewer #3: Yes

5. Is the manuscript presented in an intelligible fashion and written in standard English?

Reviewer #1: No

Reviewer #2: Yes

Reviewer #3: Yes

6. Review Comments to the Author

You may also provide optional suggestions and comments to authors that they might find helpful in planning their study.

**Reviewer #1:** Title: Women's Experiences and Outcomes of Abortion Care in Sub-Saharan Countries:

A Mixed Methods Systematic Review Protocol

I have had the opportunity to review your protocol for the systematic review on "Women's Experiences and Outcomes of Abortion Care in Sub-Saharan Countries", and I would like to provide you with some suggestions for improvement:

The protocol's overall language quality is good; there are some grammatical errors throughout the entire protocol. Before the next submission, I recommend that the entire protocol be revised again.

1. The introduction should provide a clear rationale for conducting this systematic review. Why is understanding abortion care experiences and outcomes in Sub-Saharan countries important? Consider emphasizing the public health implications.

2. Clarify the research question: The research question should be clearly defined and specific. Make sure that it is focused on a particular aspect of women's experiences and outcomes of abortion care in Sub-Saharan countries.

2. Justify the choice of study designs: Provide a rationale for why you have chosen to include both qualitative and quantitative studies in your review. Explain how each type of study will contribute to answering your research question.

3. Define inclusion and exclusion criteria: Clearly outline the criteria that will be used to select studies for inclusion in the review. This will help ensure that the selection process is transparent and reproducible.

4. Consider potential biases: Discuss potential sources of bias in the included studies, such as publication bias or selection bias. Outline how you plan to address these biases in your review.

5. Quality Assessment: Address how heterogeneity in study quality will be handled.

6. Describe data extraction and synthesis methods: Provide details on how data will be extracted from the included studies and how it will be synthesized to answer the research question. This will help readers understand the analytical approach you plan to take. Consider using a standardized form. Describe the planned synthesis methods (qualitative, quantitative, or mixed methods). Will a meta-analysis be conducted?

7. Consider implications for policy and practice: Think about how the findings of your review could inform policy and practice in Sub-Saharan countries. Consider including a discussion of potential implications in your protocol.

Overall, your protocol is well-structured and provides a solid foundation for conducting a systematic review on this important topic. By addressing these suggestions, you can strengthen the rigor and relevance of your study.

**Reviewer #2:** General comments

The over all structure of the paper is good and it is well written according to the journal guideline.

Specific comments

1. Title- does experience and outcome are related? Why the researchers want to address both outcomes as the same time?

2. Abstract- this section missed the inclusion Criteria. What is the need of discussion in the protocol?

3. Review questions- what is the purpose of the second question? Does the researcher want to synthesis the tools used to assess the abortion care experience?

4. Inclusion and exclusion criteria- the authors states that they will use population, concept and context (PCC) as indicated in the JBI (line number 91). However, the JBI manual suggests that population phenomena of interest and context (PPC). How do you respond for these inconsistencies?

5. In the table1 what are abortion outcomes related to health care interventions?

6. In the inclusion and exclusion criteria what do you think if you indicate separately for quantitative and qualitative studies?

7. Lien150- the authors indicate that they will use convergent approach. If so, what about the transformation of data from quantitative to qualitative (qualitized data)? or vice versa?

**Reviewer #3: **Overall Assessment:

The manuscript presents a well-structured and important systematic review protocol aimed at synthesizing evidence on women's abortion care experiences and outcomes in sub-Saharan Africa. The topic is highly relevant given the public health implications of abortion care in this region, where legal, cultural, and healthcare barriers significantly affect women's experiences and outcomes. The approach to integrating both qualitative and quantitative data is commendable, as it promises a comprehensive understanding of the multifactorial challenges faced by women. However, several areas could benefit from further elaboration and clarification to strengthen the protocol's robustness and potential impact.

Here are my suggestions for improving your manuscript:

1.Introduction: Consider providing a stronger justification for selecting studies published from 2010 onwards. While recent developments in abortion care practices and legal changes likely inform this choice, explicitly stating this rationale would help clarify the scope of the review. Additionally, expanding on the existing gaps in the literature regarding women's abortion care experiences and outcomes in Sub-Saharan Africa would strengthen the argument for why this systematic review is necessary.

2.Methods: Your search strategy is thorough, but it would be beneficial to address how you plan to manage potential language biases due to the exclusion of non-English studies. In the data synthesis section, please provide more detail on how you intend to integrate qualitative and quantitative data using the convergent, segregated synthesis approach. This will clarify the methodology for readers. Additionally, while the use of JBI checklists for quality assessment is appropriate, it would be helpful to explain how the quality assessment results will influence the synthesis of findings. Will studies of lower quality be weighted differently or potentially excluded from the final analysis?

3.Discussion: While you have effectively highlighted the importance of understanding abortion care experiences and outcomes, I recommend placing more emphasis on how your findings could directly influence abortion care policies and practices in Sub-Saharan Africa. Furthermore, consider outlining specific future research questions or areas that might arise from this systematic review to guide further studies in this critical area.

4.Ethical Considerations: Although you correctly state that ethical approval is waived due to the use of published data, it may be beneficial to briefly discuss the ethical implications of interpreting and using sensitive data related to abortion experiences. This consideration could add depth to your ethical discussion.

5.Author Contributions: It might be useful to specify the exact contributions of each author to enhance transparency.

6.Formatting and Clarity: Finally, ensure consistency in terminology throughout the manuscript, particularly with terms such as "woman-centred" versus "person-centred." This will enhance the clarity and professionalism of your manuscript.

7. PLOS authors have the option to publish the peer review history of their article (what does this mean?). If published, this will include your full peer review and any attached files.

Reviewer #1: **Yes: **Mesfin Abebe

Reviewer #2: No

Reviewer #3: No

---

## [Author Response · Author response to Decision Letter 0]

3 Oct 2024

PONE-D-24-05732 

Women's Experiences and Outcomes of Abortion Care in Sub-Saharan Countries: A Mixed Methods Systematic Review Protocol 

Point-by-point response to reviewers’ feedback 

We would like to thank the reviewers for sharing their constructive feedback. We have provided my point-by-point responses for each comment below in a table, and changes have been made to the manuscript accordingly.

Reviewer 1

1.The introduction should provide a clear rationale for conducting this systematic review. Why is understanding abortion care experiences and outcomes in Sub-Saharan countries important? Consider emphasizing the public health implications. 

Response: As suggested, we have now included the study's rationale in the introduction in page 3 lines 50-57 provided below as follows:

“Abortion remains a public health concern affecting women's reproductive health rights, attributing to substantial morbidity and mortality in resource-limited countries. Sub-Saharan Africa (SSA) shares high burdens of abortion-related morbidity and mortality, contributing 70% of global maternal deaths in 2020. From 2010-2014, 77% of abortions were estimated to be unsafe in SSA, while the proportions were estimated to be 45% worldwide. The disproportionate burden is associated with the lack of service availability and accessibility, restricted abortion policy, poor infrastructure, lack of skilled providers, abortion-related stigma and discrimination in the region, exacerbating abortion care experiences and outcomes.” 

Please see page 4, lines 85-95, for the detailed rationale of the study protocol.

2. Clarify the research question: The research question should be clearly defined and specific. Make sure that it is focused on a particular aspect of women's experiences and outcomes of abortion care in Sub-Saharan countries. 

Response: As we have shown in the manuscript, we have described the research questions in the review. The review questions are displayed in lines 99-100. 

3. Justify the choice of study designs: Provide a rationale for why you have chosen to include both qualitative and quantitative studies in your review. Explain how each type of study will contribute to answering your research question.

Response: Thank you. We justify why we will include qualitative and quantitative studies in the review under the study designs and cite the designs for those needing the details in page 5, lines 101-108, and we have provided the revised statements below:

“This systematic review protocol was registered in PROSPERO (ID: CRD42023461963). We will use the Joanna Briggs Institute (JBI) approach to Mixed Methods Systematic Reviews. The approach to integrating both qualitative and quantitative data will be adopted to ensure a comprehensive understanding of the multifactorial challenges women face while receiving abortion care. The Preferred Reporting Items for Systematic Reviews and Meta-Analysis Protocols (PRISMA-P) were used to define and develop the key content of a review protocol. We will use the PRISMA 2020 guideline for reporting the results of systematic reviews.”

4. Define inclusion and exclusion criteria: Clearly outline the criteria that will be used to select studies for inclusion in the review. This will help ensure that the selection process is transparent and reproducible.

Response: This review's inclusion and exclusion criteria were clearly described in lines 109-119. Particularly, our review will include both quantitative and qualitative research that assessed women's experiences in abortion care and abortion care outcomes in SSA from 1 January 2010 onwards. These are studies focusing on induced abortion care services (pre-abortion care, receipt of abortion care, and post-abortion care). Our review will exclude: i) studies published in non-English languages; ii) articles focusing on technical aspects of providing abortion care, such as studies focusing on the effectiveness of surgical or medication abortions; iii) studies focusing on the quality of abortion care, miscarriage, or spontaneous abortion care; and iv) scoping reviews, systematic reviews, case reports, case series will be excluded.

5. Consider potential biases: Discuss potential sources of bias in the included studies, such as publication bias or selection bias. Outline how you plan to address these biases in your review. 

Response: Some of the bias the reviewer mentioned are applicable in meta-analysis which we will not plan to undertake. We prefer to discuss sources of potential biases and strategies to control in review findings rather than in the protocol.

6. Quality Assessment: Address how heterogeneity in study quality will be handled. 

Response: We will undertake quality appraisal using JBI tools as displayed on page 8, lines 146-154. However, given the disparity in measurement approaches for abortion care experiences and outcomes, we will not undertake a meta-analysis but separately synthesise the qualitative and quantitative findings. 

7. Describe data extraction and synthesis methods: Provide details on how data will be extracted from the included studies and how it will be synthesized to answer the research question. This will help readers understand the analytical approach you plan to take. Consider using a standardized form. Describe the planned synthesis methods (qualitative, quantitative, or mixed methods). Will a meta-analysis be conducted? 

Response: We have already provided the details of the data extraction process under ‘data extraction’ on pages 8 and 9, lines 155-168. 

Data synthesis and integration of the included studies are described on page 9, lines 169-178, as follows: “The extracted data will be checked and summarised to address the review question. The included studies' characteristics and data related to the review questions will be organised using tables and narrative summaries under abortion care experiences, abortion care outcomes, and measurement tools for abortion care experiences and outcomes. A narrative synthesis to organise and analyse the data will be employed. Qualitative and quantitative data will be analysed separately and presented in a narrative format. The quantitative and qualitative data included in this review will then be organised according to the JBI methodology for mixed methods systematic review. Then, a convergent, segregated approach will be used to synthesise and integrate the data. This integration can occur during the interpretation of results or when drawing conclusion.”

As we have responded above, given the disparity in measurement approaches for abortion care experiences and outcomes, we will not undertake a meta-analysis but separately synthesise the qualitative and quantitative findings.

8. Consider implications for policy and practice: Think about how the findings of your review could inform policy and practice in Sub-Saharan countries. Consider including a discussion of potential implications in your protocol. Overall, your protocol is well-structured and provides a solid foundation for conducting a systematic review on this important topic. By addressing these suggestions, you can strengthen the rigor and relevance of your study.

Response: We have clarified the review finding implications for policy and practices on page 10, lines 192-197, as follows: “These will help researchers to develop standardised measurement tools for abortion care experiences and outcomes that will help policymakers, program managers, and abortion providers to monitor women's abortion care experiences and outcomes and subsequently improve the quality of abortion care in SSA. Overall, the findings of this review will identify aspects of abortion care experiences that women may encounter in facilities and care outcomes that could directly influence abortion care policies and practices in SSA.”

Reviewer 2

1. The overall structure of the paper is good and it is well written according to the journal guideline. 

Response: Thank you for the positive comment and for acknowledging we are aligned with the journal guidelines.

2. Title- does experience and outcome are related? Why the researchers want to address both outcomes as the same time?

Response: We want to see women’s experiences in healthcare facilities while receiving abortion care and abortion care outcomes because of healthcare interventions. For instance, women who encounter negative experiences such as stigma and discrimination in healthcare facilities may feel guilty, ashamed, anxious, etc, after abortion, which is an outcome.

3. Abstract- this section missed the inclusion Criteria. What is the need of discussion in the protocol?

Response: We have highlighted the review's inclusion criteria in the abstract 34-36 as follows: ‘Predetermined criteria will be used to select studies that meet the review's inclusion criteria. These include all original studies published in English languages that focussed on induced abortion care and assessed women's abortion care experiences and outcomes.’ In light of the guidelines for review protocol publications, the discussion section in the abstract is used to summarise our data synthesis and analysis plans. For reference, we have provided previously published protocol papers in the same journal:

1. Jahn A, Andersen JH, Christiansen DH, Seidler A, Dalbøge A (2023) Association between occupational exposures and chronic low back pain: Protocol for a systematic review and meta-analysis. PLoS ONE 18(5): e0285327. https://doi.org/10.1371/journal.pone.0285327

2. Amri M, Ali S, Jessiman-Perreault G, Barrett K, Bump JB (2022) Evaluating healthy cities: A scoping review protocol. PLoS ONE 17(10): e0276179. https://doi.org/10.1371/journal.pone.02761793.

3. Downes M, Welters ID, Johnston BW (2023) Study protocol: A systematic review and meta-analysis regarding the influence of coagulopathy and immune activation on new onset atrial fibrillation in patients with sepsis. PLoS ONE 18(9): e0290963. https://doi.org/10.1371/journal.pone.0290963

4. Review questions- what is the purpose of the second question? Does the researcher want to synthesis the tools used to assess the abortion care experience?

Response: Yes, this review's findings determine us to synthesise the tools for abortion care experience and outcomes in sub-Saharan Africa.

5. Inclusion and exclusion criteria- the authors states that they will use population, concept and context (PCC) as indicated in the JBI (line number 91). However, the JBI manual suggests that population phenomena of interest and context (PPC). How do you respond for these inconsistencies?

Response: We got PCC frameworks, well suitable to formulate review questions for this study than other frameworks. This is updated on page 5, lines 113-115: “The Population, Concept, and Context (PCC) framework was adopted to structure review questions and guide the search for evidence.”

 The ‘concept’ in the PCC and ‘population phenomena of interest’ in the PPC represent more or less similar idea/concept, and both (PPC and PCC) are also used in a mixed method systematic review as evidenced below:

1. Tseng LO, Newton C, Hall D, et al. Primary care family physicians' experiences with clinical integration in qualitative and mixed reviews: a systematic review protocol. BMJ Open. 2023;13(7):e067576. Published 2023 Jul 11. doi:10.1136/bmjopen-2022-06757

2. Mulyana, A. M., Rakhmawati, W., Wartakusumah, R., Fitri, S. Y. R., & Juniarti, N. (2023). The Efficacy of Internet-Based Interventions in Family-Centered Empowerment Among Children with Chronic Diseases: A Mixed-Methods Systematic Review. Journal of Multidisciplinary Healthcare, 16, 3415–3433. https://doi.org/10.2147/JMDH.S440082

3. Gizaw, Z. Public health risks related to food safety issues in the food market: a systematic literature review. Environ Health Prev Med 24, 68 (2019). https://doi.org/10.1186/s12199-019-0825-5

6. In the table1 what are abortion outcomes related to health care interventions? 

Response: We have listed abortion care outcomes because of healthcare intervention in Table 1. These include but are not limited to perceived relief and happiness after abortion, care satisfaction, perceived pain more than expected, failed medication abortion, heavy bleeding, and infection.

7. In the inclusion and exclusion criteria what do you think if you indicate separately for quantitative and qualitative studies?

Response: We have fundamental inclusion and exclusion criteria for both quantitative and qualitative studies. Separating eligibility criteria for quantitative and qualitative studies is unnecessary for this review.

8. Lien150- the authors indicate that they will use convergent approach. If so, what about the transformation of data from quantitative to qualitative (qualitized data)? or vice versa?

Response: Our review questions focus on different dimensions of a phenomenon, which must follow a convergent segregated approach to its synthesis and integration. No data transformation will take place for this review. As clearly described in the JBI Manual, we will use a convergent segregated approach (not convergent integrated), which involves the independent synthesis of quantitative data and qualitative data, leading to the generation of quantitative evidence and qualitative evidence, which will then be integrated at the interpretation phase: 

1.Lucylynn Lizarondo, C.S., Judith Carrier, Christina Godfrey, Kendra Rieger, Susan Salmond, Joao Apostolo, Pamela Kirkpatrick, Heather Loveda, Chapter 8: Mixed methods systematic reviews (2020), JBI Manual for Evidence Synthesis., L.C. Aromataris E, Porritt K, Pilla B, Jordan Z, Editor. 2024.

Reviewer 3

1 Introduction: Consider providing a stronger justification for selecting studies published from 2010 onwards. While recent developments in abortion care practices and legal changes likely inform this choice, explicitly stating this rationale would help clarify the scope of the review. Additionally, expanding on the existing gaps in the literature regarding women's abortion care experiences and outcomes in Sub-Saharan Africa would strengthen the argument for why this systematic review is necessary. 

Response: Thank you. We have included justification for selecting studies published from 2010 onwards on page 4, lines 75-84, as follows: “The Maputo protocol, which emphasised equality and women's reproductive health rights in Africa, was adopted in 2003 and effective in 2005, and a total of 43 African nations signed the protocol in 2007. Maputo protocol article 14 described women's sexual and reproductive health rights, authorising women to access abortion services in cases of sexual violence, rape, incest, and where the continued pregnancy endangers the life of the mother or the foetus. This period was the time when some sub-Saharan countries started reforming their abortion policies, developing abortion policies implementation guidelines and expanding access to safe abortion services. Our systematic review will be focused on studies published from 2010 onwards, seeing the recent developments in abortion care practices and legal changes that influence abortion care experiences and outcomes in SSA.” We have also expanded the existing gaps in the literature regarding women's abortion care experiences and outcomes in Sub-Saharan Africa in the introduction section.

2. Methods: Your search strategy is thorough, but it would be beneficial to address how you plan to manage potential language biases due to the exclusion of non-English studies. 

Response: We anticipate that most available studies will be published in English-language journals. However, to provide context, we will record the number of non-English studies excluded during screening and highlight this as a limitation.

3. In the data synthesis section, please provide more detail on how you intend to integrate qualitative and quantitative data using the convergent, segregated synthesis approach. This will clarify the methodology for readers. 

Response: We have clarified the data synthesis and cited the data synthesis and integration of the JBI methodology for mixed methods systematic review in the manuscript on page 9, lines 169-178: “The extracted data will be checked and summarised to address the review quest

---

## [Decision Letter · Decision Letter 1]

17 Jan 2025

Women's Experiences and Outcomes of Abortion Care in Sub-Saharan Countries: A Mixed Methods Systematic Review Protocol

PONE-D-24-05732R1

Dear Dr. Wakgari,

We’re pleased to inform you that your manuscript has been judged scientifically suitable for publication and will be formally accepted for publication once it meets all outstanding technical requirements.

Kind regards,

Patrick Goymer

Staff Editor

PLOS ONE

Additional Editor Comments (optional):

Reviewers' comments:

Reviewer's Responses to Questions

**Comments to the Author**

1. Does the manuscript provide a valid rationale for the proposed study, with clearly identified and justified research questions?

Reviewer #1: Yes

Reviewer #3: Yes

2. Is the protocol technically sound and planned in a manner that will lead to a meaningful outcome and allow testing the stated hypotheses?

Reviewer #1: Yes

Reviewer #3: Yes

3. Is the methodology feasible and described in sufficient detail to allow the work to be replicable?

Reviewer #1: Yes

Reviewer #3: Yes

4. Have the authors described where all data underlying the findings will be made available when the study is complete?

Reviewer #1: Yes

Reviewer #3: Yes

5. Is the manuscript presented in an intelligible fashion and written in standard English?

Reviewer #1: Yes

Reviewer #3: Yes

6. Review Comments to the Author

You may also provide optional suggestions and comments to authors that they might find helpful in planning their study.

Reviewer #1: The author has thoroughly addressed all of my previous comments. They have carefully considered each point I raised and provided comprehensive responses or made the necessary revisions to the manuscript. The changes made reflect a clear understanding of the feedback provided, and the author has effectively incorporated these suggestions to enhance the overall quality and clarity of the work.

Reviewer #3: All the comments and suggestions I mentioned during the review is addressed by Authors. I have no further questions or comments for them.

7. PLOS authors have the option to publish the peer review history of their article (what does this mean?). If published, this will include your full peer review and any attached files.

Reviewer #1: **Yes: **Mesfin Abebe

Reviewer #3: No

---

## [Editor Report · Acceptance letter]

20 Jan 2025

PONE-D-24-05732R1 

PLOS ONE

Dear Dr. Wakgari, 

I'm pleased to inform you that your manuscript has been deemed suitable for publication in PLOS ONE. Congratulations! Your manuscript is now being handed over to our production team.

Kind regards, 

on behalf of

Dr Patrick Goymer 

Staff Editor

PLOS ONE